# Analyzing Lottery Ticket Hypothesis from PAC-Bayesian Theory Perspective

**Keitaro Sakamoto**
The University of Tokyo
sakakei-1999@g.ecc.u-tokyo.ac.jp

**Issei Sato**
The University of Tokyo
sato@g.ecc.u-tokyo.ac.jp

## Abstract

The lottery ticket hypothesis (LTH) has attracted attention because it can explain why over-parameterized models often show high generalization ability. It is known that when we use iterative magnitude pruning (IMP), which is an algorithm to find sparse networks with high generalization ability that can be trained from the initial weights independently, called *winning tickets*, the initial large learning rate does not work well in deep neural networks such as ResNet. However, since the initial large learning rate generally helps the optimizer to converge to flatter minima, we hypothesize that the winning tickets have relatively sharp minima, which is considered a disadvantage in terms of generalization ability. In this paper, we confirm this hypothesis and show that the PAC-Bayesian theory can provide an explicit understanding of the relationship between LTH and generalization behavior. On the basis of our experimental findings that IMP with a small learning rate finds relatively sharp minima and that the distance from the initial weights is deeply involved in winning tickets, we offer the PAC-Bayes bound using a spike-and-slab distribution to analyze winning tickets. Finally, we revisit existing algorithms for finding winning tickets from a PAC-Bayesian perspective and provide new insights into these methods.

## 1 Introduction

The high generalization ability of modern neural networks can be attributed to the heavier overparameterization and effective learning algorithms [22, 30, 41]. This increase in the number of parameters leads to high computational cost and high memory usage, and network pruning is one of the effective techniques for addressing these problems [12, 14, 23]. After pruning a significant number of parameters, the pruned network can often work well with little or no accuracy loss. However, training this sparse subnetwork independently from the initial weights often does not work, and we can only obtain the sparse subnetwork through pruning after training the whole network.

Frankle and Carbin [10] presented "Lottery Ticket Hypothesis (LTH)" that states the existence of winning tickets: small but critical subnetworks which can be trained independently from scratch. They proposed an algorithm called iterative magnitude pruning (IMP) to obtain a winning ticket. They pointed out that, for deeper networks such as VGG [35] and ResNet [16], small learning rate is required to obtain a winning ticket. However, since a large learning rate helps the generalization ability of neural networks [25], the learning rate should be well controlled to find a winning ticket that has a higher test accuracy. Frankle et al. [11] found a correlation between the stability to SGD noise and the ability to find the winning ticket and empirically showed that the large learning rate moves weights too much under the low-stability learning process.

In this paper, we first empirically show that winning tickets are actually more vulnerable to label noise setting compared to the subnetwork created with the large learning rate; that is, the generalization ability of winning tickets is degraded due to the learning rate constraint. In this connection, we then

36th Conference on Neural Information Processing Systems (NeurIPS 2022).

focus on the two concepts flatness and the distance from the initial weights of the winning tickets. We next apply the PAC-Bayesian theory to LTH on the basis of the flatness motivation and show that it can explain the relationship between LTH and generalization behavior. We use the PAC-Bayes bound for a spike-and-slab distribution to analyze winning tickets, which is based on our experimental findings that reducing the expected sharpness restricted to an unpruned parameter space and adding the regularization of distance from the initial weights can enhance the test performance of winning tickets. Finally, we revisit existing algorithms such as IMP, continuous sparsification [34] from the point of view of the PAC-Bayes bound optimization. This consideration gives an interpretation of these methods as an approximation of bound optimization.

To sum up, our contributions are as follows.

- We experimentally show that IMP with a small learning rate finds relatively sharper minima and that the distance from the initial weights is critical for IMP, i.e., balancing the distance and the training error helps to find them.

- On the basis of our findings, we reveal that the PAC-Bayesian formulation for a spike-and-slab distribution effectively captures the winning tickets behavior.

- We revisit the existing algorithms from the PAC-Bayesian perspective and explain their behavior.

## 2 Related Work

**Learning Rate**     The initial large learning rate often improves generalization of deep neural networks [19, 25]. The large learning rate generally helps an optimizer to converge to flatter minima [24, 38], and these flatter minima are advantageous for generalization ability rather than sharper minima [18, 20, 21]. The relationship between flatness and generalization can be considered from a PAC-Bayesian perspective [5, 31, 32] (see Appendix A). Frankle et al. [11] provided the insights in why IMP with the large learning rate fails into find winning tickets in some problem settings. They proposed IMP with rewinding to an early epoch to avoid the very early training process because training is not stable to SGD noise in the early training regime (see Appendix A).

**Empirical Results on LTH**     Some studies have also investigated the flatness and the distance from the initial weights on LTH. Bain [2] plotted the loss landscape of winning tickets visually and found that IMP produces more convex and sharper minimum relative to random pruning. Bartoldson et al. [3] found that pruning regularizes similarly to noise injection, and they discussed the generalization of pruned networks considering flatness. They measured flatness by using the trace of Hessian and gradient covariance matrix. He et al. [17] refers the distance from the initial weights to analyze the label noise robustness of winning tickets; they stated that the influence of the label noise is lessened if the distance is suppressed. Liu et al. [26] discussed the relationship between winning tickets and the learning rate considering the similarity between initial and trained weights in terms of pruning mask overlap instead of the distance from the initial weights. There is a recent study on finding a mask that shows good accuracy without training weights at all [33, 43].

**Theoretical Results on LTH**     Malach et al. [29] demonstrated a subnetwork with a comparable accuracy to the original network in a sufficiently over-parameterized network, without any training. Zhang et al. [42] analyzed the winning tickets generalization based on sample complexity. Although it is limited to the case of a one-hidden-layer neural network, they provided insight into why the sparsity increases the generalization ability. For a PAC-Bayesian theory, Hayou et al. [15] also used spike-and-slab prior and posterior distributions. However, their motivation and purpose are completely different from our work. Their aim was to obtain a sophisticated pruning mask by optimizing the PAC-Bayes bound, and this is mainly in the context of network pruning rather than LTH. Our work differs in that we use it to analyze the generalization behavior of a given winning ticket on the basis of our empirical findings on the learning rate, the flatness, and the distance from the initial weights. In addition to these, we also discuss the relationship with existing algorithms.

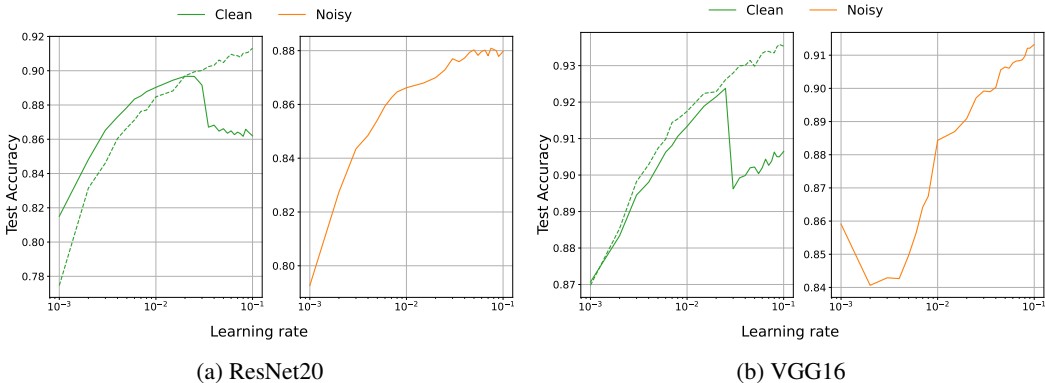

Figure 1: Test accuracy on CIFAR10 whose labels are randomly flipped (clean: green line, noisy: orange line) when ResNet20 (90% sparse) and VGG16 (99% sparse) are trained. These subnetworks are produced by IMP with various learning rates. The dashed green line shows the unpruned baseline with the same learning rate. In the label noise setting, the model has not yet converged at the same number of epochs as the clean setting; therefore we train them until convergence, and training epoch is increased from the setting of Frankle and Carbin [10].

## 3 Empirical Analysis

We first show some empirical results because our new findings about winning tickets in this empirical analysis motivate the PAC-Bayesian analysis for LTH; thus, we interpret the results on the basis of the PAC-Bayesian perspective in the next section.

We empirically investigated the properties of winning tickets mainly related to the learning rate. The small learning rate is good for finding the winning ticket, which is contrary to the intuition that a large learning rate is good in terms of generalization ability. First, we show the small learning rate is actually disadvantageous for generalization ability of subnetworks through experiments under label noise. Next, as a reason for this, we show that the small learning rate finds the relatively sharper minima, and it is possible to find the winning tickets in the flatter minima using sharpness regularization. We also focused on the distance from the initial weights as a reason for why the small learning rate is important for IMP. On the basis of these findings, we will develop a discussion from a PAC-Bayesian perspective in the next section. We will show this perspective will theoretically support the findings in this section. We followed the experimental setting of Frankle and Carbin [10] and used a modified version of OpenLTH repository [9].

### 3.1 Vulnerability to label noise

We examined the test accuracy when some fractions of the labels in the training set are randomly flipped to see whether the generalization behavior of winning tickets is degraded by the constraint that the learning rate must be small. Figure 1 shows the test accuracy on clean and label noise datasets of sparse subnetworks produced by IMP with different learning rates. As for the no label noise setting (green line), there is an accuracy drop at some point as the learning rate is increased. We added the original unpruned baseline (dashed green line) to discuss if the subnetwork is a winning ticket, and this baseline shows no such accuracy drop when increasing the learning rate. The subnetwork eventually performs worse than the original unpruned network, which means that IMP ultimately fails to find the winning ticket. In contrast, the test accuracy generally continues to improve without decreasing as the learning rate is increased in the high label noise setting (orange line). The test accuracy increases even when it is not a winning ticket for no label setting at the same learning rate. To sum up, the large learning rate is not suitable for finding winning tickets under a clean dataset but is advantageous in the high label noise setting.

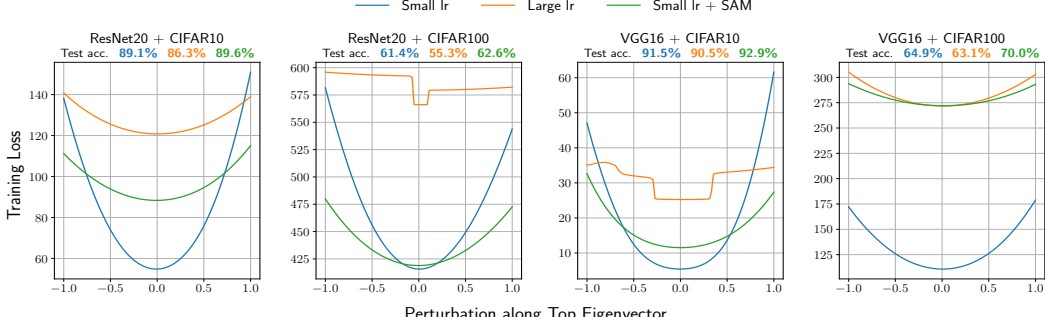

Figure 2: 1-d training loss landscape along eigenvector corresponding to largest eigenvalue of Hessian around trained parameters. The test accuracies in the caption correspond to the parameter at the center of perturbation (0.0 on the horizontal axis) and show accuracy for the small learning rate (blue), the large learning rate (orange), and the small learning rate + SAM (green) form left to right. The subnetwork of ResNet20 has 90% sparsity, and that of VGG16 has 99% sparsity. The top eigenvectors are calculated by using PyHessian [40].

## 3.2 Flatness

Now that we have seen that winning tickets have undesirable properties due to the small learning rate, we investigated whether this result comes from the difference in flatness around the found solution. There are many previous studies that discuss the relationship between the learning rate and flatness [24, 38]. In this experiment, perturbations are added only to unpruned weights to consider flatness on pruned networks. We will discuss this justification in detail in Section 4.

First, we visualized the loss landscape shape of subnetworks produced by IMP. Figure 2 shows the 1-d loss landscape of the subnetwork that has a high sparsity. This landscape shows the training loss around the trained weights adding perturbation restricted to unpruned parameters. We can see that the parameters of the winning ticket (small learning rate) are in the sharper minimum compared with the large learning rate. The large learning rate can find a flatter minimum; however, the training loss is higher and the test accuracy is worse than the small learning rate. This graph is a fixed sparsity loss landscape; therefore, it is not possible to discuss whether or not the subnetwork is a winning ticket from this graph. There is actually an accuracy drop in the large learning rate setting when sparsity is changed, which means this is not a winning ticket.

Given that IMP requires a small learning rate rather than a large one, these results suggest two possible interpretations; 1) sharp minimum is essential to find winning tickets, therefore the large learning rate fails in IMP, 2) sharp minimum is simply the result of small learning rate training, and flat minimum is better for winning tickets if possible. To investigate these possibilities, we used sharpness-aware minimization (SAM) [8] and neural variable risk minimization (NVRM) [37] to search for parameters that lie in neighborhoods having uniformly low loss. They differ in that SAM minimizes the maximum loss in the neighborhood, whereas NVRM minimizes the expected loss in the neighborhood. We used both of them to compare with normal SGD as a baseline. These optimizers are based on SGD with the same setting as the original LTH paper [10], and the noise considered in these methods is limited to the unpruned parameters. Figure 2 also shows the loss landscape when SAM is used; It can actually reach a flatter minimum. The loss landscape is as flat as the large learning rate setting; however, unlike this, the test accuracy is higher than that of the small learning rate.

Next, we investigated the trace of Hessian to analyze the flatness of winning tickets created by these optimizers. The trace of Hessian is used as a measure of flatness in the prior work [4, 21, 39]. Table 1 shows that SGD with the small learning rate finds relatively a sharper minimum and that IMP can find a flatter one by using SAM or NVRM. These optimizers can find relatively flatter minima, which improves the test accuracy to some extent compared with the SGD with the small learning rate (see Appendix B.1).

We also found that the large learning rate still cannot find winning tickets even though we use SAM instead of SGD (see Appendix B.2). The large learning rate has already found relatively flatter

Table 1: Trace of Hessian for ResNet20 and VGG16 trained on CIFAR10 and CIFAR100. We used three optimizers, SAM, NVRM, and SGD. The hyperparameter of SAM $\rho$ and NVRM $b$ are chosen from $\{0.05, 0.1, 0.2, 0.5\}$ and $\{0.014, 0.018, 0.022, 0.026\}$ respectively with the highest test accuracy. As a baseline, we show the results of SGD with a small learning rate, and the learning rate is also set to small for SAM and NVRM. The trace of Hessian is calculated by PyHessian [40]. We averaged over three different subnetworks.

| Dataset | Sparsity (%) | ResNet20 | | | VGG16 | | |
|---|---|---|---|---|---|---|---|
| | | SGD | SAM | NVRM | SGD | SAM | NVRM |
| CIFAR10 | 90 | 1556 | 365 | 1238 | 107 | 67 | 70 |
| | 95 | 1786 | 791 | 1218 | 171 | 79 | 62 |
| CIFAR100 | 90 | 4010 | 2071 | 3190 | 231 | 119 | 159 |
| | 95 | 6340 | 758 | 2997 | 411 | 226 | 190 |

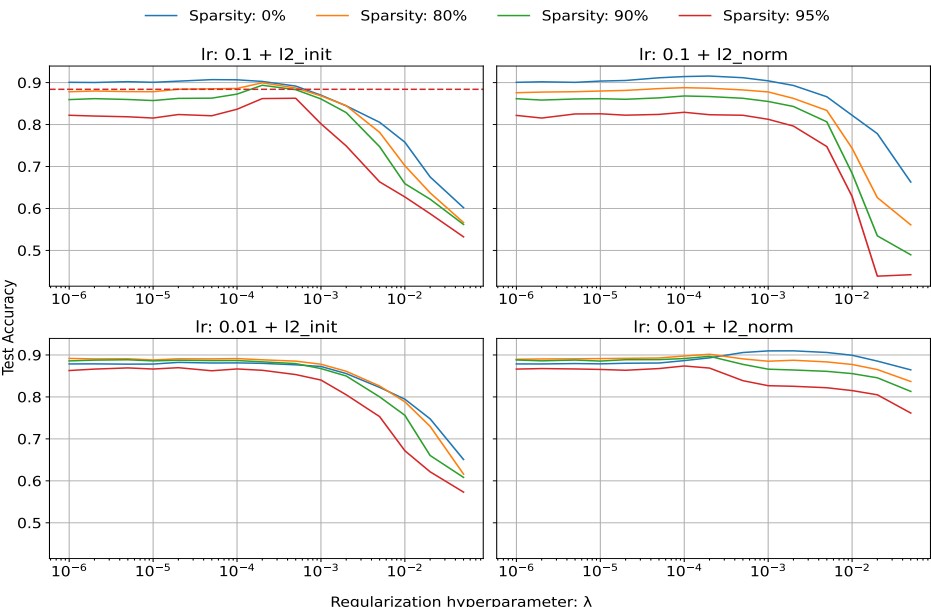

Figure 3: Test accuracy of ResNet20 trained on CIFAR10 when regularization term is added. Top row shows the results of the large learning rate (0.1) and bottom row shows those of the small learning rate (0.01). Left: $l2\_init$, Right: $l2\_norm$. The dashed red line in the top left shows the unpruned baseline with the small learning rate (0.01).

minima as shown in Figure 2; therefore, it is considered that the results do not change by using SAM. This fact implies that the flatness around the parameters found by SGD with a large learning rate and by SAM with a small learning rate are similar, however winning tickets cannot be found with the large learning rate because they find different solutions in terms of some other properties. Next, we analyze this difference by focusing on the distance from the initial weights.

### 3.3 Distance from initial weights

As a reason for the small learning rate constraint, we hypothesized that winning tickets can only be found by IMP within a range not far from the initial weights. In order to confirm this, we ran IMP suppressing this distance and compare it with the usual regularization by the $L_2$ norm regularization. In addition, we also discussed the pruning mask structure in Appendix B.3 in relation to the distance from the initial weights.

We empirically confirmed that the winning ticket can be obtained via the regularization of this distance even under the large learning rate setting. Let $\mathcal{L}_{\mathcal{S}}$ be the empirical risk on the training samples $\mathcal{S}$,

$\lambda$ be a regularization hyperparameter, $\boldsymbol{\theta}$ be network weights, and $\boldsymbol{m} \in \{0,1\}^{|\boldsymbol{\theta}|}$ be a pruning mask. In this experiment, the parameters were regularized only for unpruned weights. Specifically, we designed the loss function with $l2\_init$ and $l2\_norm$, respectively, as follows.

$$\mathcal{L}_{\mathcal{S}}(\boldsymbol{\theta}) + \lambda \|\boldsymbol{m} \odot (\boldsymbol{\theta} - \boldsymbol{\theta}_{\text{init}})\|_2^2; \quad \mathcal{L}_{\mathcal{S}}(\boldsymbol{\theta}) + \lambda \|\boldsymbol{m} \odot \boldsymbol{\theta}\|_2^2. \tag{1}$$

Figure 3 shows the test accuracy with different regularizations by changing the hyperparameter $\lambda$. We plot 80%, 90% and 95% sparsity subnetworks and the original unpruned network as a baseline. If the subnetwork accuracy is close to the whole network accuracy, it is considered to be successful in finding the winning ticket. As discussed previously, the large learning rate fails to find winning tickets unlike the small learning rate setting; however adding the regularization from the initial weights changes this trend. In Figure 3, $\lambda$ around $2 \times 10^{-4}$ shows that a winning ticket is found since the accuracy drop from the whole network is suddenly reduced, and there is no such a trend when $L_2$ norm regularization is added. This means that IMP can obtain winning tickets suppressing the distance from the initial weights even with the large learning rate. This finding is shown more clearly in Appendix B.4. We confirmed it by examining the test accuracy when sparsity is changed for problem settings other than ResNet20 + CIFAR10.

We can also obtain interesting results when the learning rate is small. The accuracy drop is small while $\lambda$ is small; however it becomes large increasing $\lambda$. In the case of $l2\_init$, it is possible that the gap widens because sparse networks are more affected by strong regularization (it is the same for large learning rate setting), but in the case of $l2\_norm$, the gap widens significantly even though generalization ability increases because of the regularization. The strong norm regularization makes IMP fail to find a winning ticket even if a winning ticket could be found originally.

These results are related to the prior-mean selection from the PAC-Bayesian perspective. In the training of a normal network, there is a trade-off between suppressing the parameter norm, i.e., reducing the KL term, and reducing the training loss. Therefore, it is important to ensure a balance between them. As for IMP, there is a specific situation where suppressing the parameter norm makes the training loss large due to the failure to find a winning ticket. If we take the initial weights as a prior mean instead of 0, the training loss can decrease by suppressing the KL term when the trained weights are far from the initial weights. This experiment corresponds to what to take as a prior mean in terms of minimizing PAC-Bayes bound, and this result indicates that setting the initial weights to a prior mean seems to be compatible with minimizing the PAC-Bayes bound in the case of winning tickets.

## 4 PAC Bayesian Analysis on LTH

First, we present some possible definitions of subnetwork flatness and consider a PAC-Bayes bound of a spike-and-slab formulation based on our experiments. Next, we show that this bound captures the generalization behavior of winning tickets and revisit existing algorithms from the perspective of this bound.

### 4.1 Flatness on winning tickets

While we could simply consider the neighborhood of the trained weights in an unpruned neural network, it is not trivial to define the flatness of pruned networks depending on how the pruned weights are taken into consideration. The possible measure of the expected sharpness are as follows.

1. Add noise to the parameters restricted on unpruned parameter space.
2. Add noise to the parameters including the pruned weights.
3. Recover pruned weights and train the whole network to convergence (re-dense training [13]) and measure its flatness.

Measure 2 is the same as the unpruned neural network, but the sparse weights in the whole parameter space can no longer be in the local minimum; thus, it is uncertain whether flatness has any meaning in such a setting. He et al. [17] conducted re-dense training and showed that solutions at high sparsity are no longer minimizers in high dimensions. They also found that the winning ticket has higher sharpness than the original network based on Measure 3 and concluded that highly sparse solutions do not stick around the flat basins of minimizers. However, none of these metrics has any justification.

It is known that flatness can be viewed from a PAC-Bayesian perspective [5, 31, 32]. We use a PAC-Bayes bound of a spike-and-slab formulation, where expected sharpness corresponds to Measure 1. We will also compare this formulation with the normal Gaussian formulation (Measure 2) through numerical experiments and show that the spike-and-slab formulation captures the generalization behavior of winning tickets better. This supports our findings that using SAM and NVRM on pruned networks can enhance the generalization accuracy of winning tickets.

## 4.2 Spike-and-slab formulation

There are several problems when it comes to using the Gaussian distribution as prior and posterior for analyzing winning ticket properties. As discussed in Section 3.3, the distance from the initial weights of the winning ticket is expected to be small. We set the prior mean as the initial weights to take advantage of this property; however the original PAC Bayesian formulation based on the Gaussian distribution has the disadvantage that the pruned weights have weight $0$ and the norm of the initial weights corresponding to these weights remain in the KL part. This not only results in a large bound but also behaves contrary to the purpose of getting sparse subnetworks when the bound is optimized. This is because the more sparse the subnetwork is, the larger the bound becomes. In addition, noise is inevitably added to the pruned weights when considering expected sharpness since the variance of the pruned part cannot be set to zero.

In order to limit the distance from the initial weights and noise added in expected sharpness only to the unpruned weights, we use a spike-and-slab distribution, which is the mixture of the Gaussian distribution $\mathcal{N}$ and Dirac delta distribution with a peak at zero-weight $\delta_{\{0\}}$, in the PAC-Bayes bound. Let $\boldsymbol{\sigma}_p$ and $\boldsymbol{\sigma}_q$ be vectors whose $i^{\text{th}}$ element is a variance of Gaussian distribution, $\boldsymbol{\lambda}_p$ and $\boldsymbol{\lambda}_q$ be vectors that represents the mixture ratio of prior and posterior, respectively. $\boldsymbol{\theta}$ represents the network parameter, and $\boldsymbol{\theta}_{\text{init}}$ is the initial weights and $\bar{\boldsymbol{\theta}}$ is the trained weights.

We design the prior $\mathbb{P}$ and posterior $\mathbb{Q}$ as follows.

$$
\begin{aligned}
\mathbb{P}(\theta_i) &= (1 - \lambda_{p,i})\delta_{\{0\}} + \lambda_{p,i}\mathcal{N}(\theta_i \mid \theta_{\text{init},i}, \sigma_{p,i}), \\
\mathbb{Q}(\theta_i) &= (1 - \lambda_{q,i})\delta_{\{0\}} + \lambda_{q,i}\mathcal{N}(\theta_i \mid \bar{\theta}_i, \sigma_{q,i}).
\end{aligned}
\tag{2}
$$

The KL divergence of the spike-and-slab distribution [36] can be calculated as

$$
\begin{aligned}
&\text{KL}\left[\mathbb{Q}(\boldsymbol{\theta})\|\mathbb{P}(\boldsymbol{\theta}))\right] \\
&= \sum_i \left(\lambda_{q,i}\left(\log\frac{\sigma_{p,i}}{\sigma_{q,i}} + \frac{\sigma_{q,i}^2 + (\bar{\theta}_i - \theta_{\text{init},i})^2}{2\sigma_{p,i}^2} - \frac{1}{2}\right) + \text{kl}\left[\lambda_{q,i}\|\lambda_{p,i}\right]\right),
\end{aligned}
\tag{3}
$$

where

$$
\text{kl}\left[\lambda_{q,i}\|\lambda_{p,i}\right] = \lambda_{q,i}\log\frac{\lambda_{q,i}}{\lambda_{p,i}} + (1 - \lambda_{q,i})\log\left(\frac{1 - \lambda_{q,i}}{1 - \lambda_{p,i}}\right).
\tag{4}
$$

Here, $\lambda_{p,i}$ is set to the target sparsity, so if we want 90% sparsity winning tickets, then $\lambda_{p,i}$ is set to 0.1. Since the structure of the pruned network is given, we set the element of $\lambda_{q,i}$ to 0 or 1 asymptotically according to the pruning mask $\boldsymbol{m}$ and obtain the following kl divergence about $\lambda$. This operation has been conventionally done in the entropy discussion [28].

$$
\text{kl}\left[\lambda_{q,i}\|\lambda_{p,i}\right] = \begin{cases} -\log\lambda_{p,i} & \text{(unpruned)} \\ -\log(1 - \lambda_{p,i}) & \text{(pruned)} \end{cases}.
\tag{5}
$$

The expected sharpness is as follows.

$$
\mathcal{L}_{\mathcal{S}}(\mathbb{Q}(\boldsymbol{\theta})) - \mathcal{L}_{\mathcal{S}}(\bar{\boldsymbol{\theta}}) = \mathbb{E}_{\boldsymbol{\epsilon}}\left[\mathcal{L}_{\mathcal{S}}(\bar{\boldsymbol{\theta}} + \boldsymbol{\epsilon})) - \mathcal{L}_{\mathcal{S}}(\bar{\boldsymbol{\theta}})\right],
$$

where

$$
\epsilon_i \sim \begin{cases} \mathcal{N}(x \mid 0, \sigma_{q,i}) & \text{(unpruned)} \\ \delta_{\{0\}} & \text{(pruned)} \end{cases}.
$$

This means that flatness of pruned networks can be discussed adding noise to only unpruned weights. The advantage of this definition compared to the Gaussian distribution setting, where we cannot avoid adding noise to pruned weights, will be discussed in the next subsection.

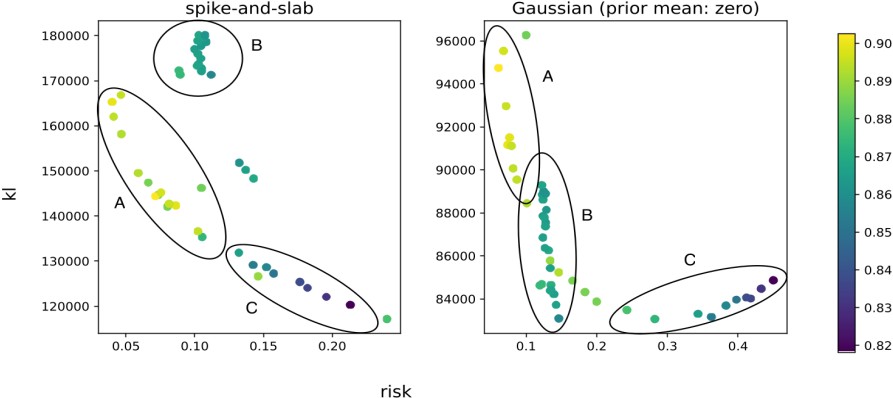

Figure 4: Correlation between training risk term and KL term when PAC-Bayes bound is optimized. We plot over subnetworks (90% sparse) generated by different learning rates of IMP (ResNet20 + CIFAR10). Left: spike-and-slab distribution, Right: Gaussian distribution with zero-mean prior.

## 4.3 Numerical Experiment

We conducted numerical experiments to confirm that our PAC-Bayes formulation can adequately explain the behavior of winning tickets. We optimized the posterior variance to minimize the PAC-Bayes bound, and plot KL term and the training loss on the posterior distribution to investigate that the bound can capture the test accuracy of the subnetworks produced by IMP. As a comparison, we also experiment with the Gaussian distribution setting using the zero-mean prior. The PAC-Bayesian bound used here is the variational KL bound [6] (Theorem D.2, and see also Appendix B.7), and we implemented the code following Dziugaite et al. [7].

Figure 4 shows the distribution of the training risk term and KL term when optimizing the PAC-Bayes bound, and the actual test accuracy is colored. We show that the bound with the spike-and-slab formulation successfully explains the behavior of winning tickets, dividing the point cloud into three groups: A) winning tickets (moderate learning rate), B) subnetworks that failed to find winning tickets (too high learning rate) and C) not much trained subnetworks (too small learning rate).

In the left figure, as the learning rate is increased, the distance from the initial weights increases and the training risk gradually decreases from C to A; The same trend also appears on the right figure. On the other hand, the distribution of B differs greatly between left and right. The right figure does not capture the test accuracy well because B should have higher test accuracy considering its KL term and training risk term. This means that, as the learning rate is increased and winning tickets can no longer be found, the KL term when prior mean is zero becomes much smaller than that of A. This is because many parameters become close to zero rather than near their initial weights when the winning ticket fails to be found (see Appendix B.5). In terms of not only the bound optimization but also the analysis of the existing winning ticket, it is preferable to use the spike-and-slab distribution to set the prior mean as the initial weights.

## 4.4 Revisiting existing algorithms

We reconsider the existing algorithms for winning ticket searching from the perspective of optimizing the PAC-Bayes bound. Although we focus only on IMP and continuous sparsification, this view could be helpful for other methods as well.

### 4.4.1 IMP

IMP is a heuristic method and does not have an explicit target function. Here, we explain how IMP behaves in the sense of a PAC-Bayes bound with our formulation instead of viewing IMP as a direct bound optimization problem.

The risk term and KL term in the PAC-Bayes bound are basically in a trade-off relationship; choosing a complex model to fit the training data will increase the KL term, while choosing a simple model

Table 2: Drop of training accuracy with different pruning criteria. We averaged the accuracy drop of five winning tickets (90% sparsity) produced by different learning rate (ResNet20, CIFAR10).

| Criteria | After pruning (%) | After rewinding (%) | After retraining (%) |
|---|---|---|---|
| large_final | −19.8 | −88.6 | −0.3 |
| small_final | −89.7 | −89.5 | −8.0 |
| random | −88.8 | −88.6 | −5.7 |

may not have a high accuracy on the training data. We point out that the two steps of IMP: 1) train the subnetwork, 2) prune the subnetwork and revert its weights, optimize the overall bound by alternately reducing one term while suppressing the increase in the other term.

**In the first step**, IMP trains the subnetwork from initial weights under a given pruning mask. This process reduces the training risk, and the increase in KL term is not expected to be so large in our formulation because the prior mean is set to the initial weights and the trained weights of IMP are not far from the initial weights as shown in Sections 3.3 and 4.3.

**In the second step**, IMP prunes a certain percentage of the smallest magnitude weights and reverts the trained weights to the initial state. Reverting $\boldsymbol{\theta}$ to $\boldsymbol{\theta}_{\text{init}}$ and changing part of $\lambda_q$ from 1 to 0 make the KL term small. The number of Gaussian KL summations decreases and the distance from initialization gets to 0, and the $\lambda$ KL part is also reduced if the prior mixture ratio $\lambda_p$ is set to the final target sparsity. This KL reduction is not dependent on the pruning criterion. The problem here is how to minimize the increase in training loss, which is related to what heuristic pruning criterion we choose and why pruning weights with a small absolute value works well.

Table 2 lists the training accuracy drop when we use three different pruning criteria; $large\_final$ leaves a large absolute value of weights and corresponds to IMP, $small\_final$ conversely leaves small weights, and $random$ prunes randomly. This notation of criteria follows that of Zhou et al. [43]. As expected, $large\_final$ has a smaller drop in training accuracy after pruning than the others. Training accuracy gets lower after reverting; If we assume that retraining can reach weights that show the same or better accuracy because weights achieving good accuracy with the same structure exist, it seems to make sense to use $large\_final$ to decrease the KL term while suppressing the increase in training loss. The results in Table 2 confirm this assumption empirically.

We can also discuss the reason why $large\_final$ pruning criterion is good for IMP by considering the following simple Taylor expansion (see Appendix B.6).

$$|f(\boldsymbol{\theta} + \Delta\boldsymbol{\theta}) - f(\boldsymbol{\theta})| \leq \frac{1}{2}\|\Delta\boldsymbol{\theta}\|_2^2 \sup_{\gamma \in [0,1]} \lambda_{\max}^{\boldsymbol{\theta} + \gamma\Delta\boldsymbol{\theta}}, \tag{6}$$

where $\lambda_{\max}^{\boldsymbol{\theta} + \gamma\Delta\boldsymbol{\theta}}$ is a top eigenvalue of Hessian $H(\boldsymbol{\theta} + \gamma\Delta\boldsymbol{\theta})$.

This provides a brief insight into why IMP succeeds by pruning small magnitude weights under the assumption that the maximum eigenvalues are not very different.

### 4.4.2 Continuous Sparsification

Continuous sparsification [34] is a method to find winning tickets by removing the parameters continuously instead of alternating between training and pruning. This target function is as follows,

$$\min_{\boldsymbol{\theta} \in \mathbb{R}^d, \boldsymbol{m} \in \{0,1\}^d} \mathcal{L}_{\mathcal{S}}(\boldsymbol{m} \odot \boldsymbol{\theta}) + \eta \cdot \|\boldsymbol{m}\|_1, \tag{7}$$

where $\eta > 0$ is a hyperparameter. Continuous sparsification is formulated as the training loss minimization with the $L_0$ regularization of weights, and a sigmoid function $\sigma$ is used for the continuous relaxation of the regularization term as follows.

$$\min_{\boldsymbol{\theta} \in \mathbb{R}^d, \boldsymbol{s} \in \mathbb{R}_{\neq 0}^d} \lim_{\beta \to \infty} \mathcal{L}_{\mathcal{S}}(\sigma(\beta\boldsymbol{s}) \odot \boldsymbol{\theta}) + \eta \cdot \|\sigma(\beta\boldsymbol{s})\|_1. \tag{8}$$

We can regard this function as an approximation of the PAC-Bayes bound of our formulation. Let $\phi_i \geq 0$ be the Gaussian KL part in 3, the summation of training risk and KL is as follows.

$$\mathcal{L}_{\mathcal{S}}(\mathbb{Q}) + \sum_i \phi_i \lambda_{q,i} + \sum_i \text{kl}\left[\lambda_{q,i} \| \lambda_{p,i}\right]. \tag{9}$$

We make three approximations: 1) replace $\mathbb{Q}$ with $\mathbb{E}[\boldsymbol{\theta}]$ over the spike-and-slab distribution by first-order Taylor expansion on the training risk, 2) simplify the second term to the $L_1$ norm of $\boldsymbol{\lambda_q}$ because the second term can be viewed as a weighted summation of $\lambda_{q,i}$, and 3) remove the third term, which can be regarded as a regularization to the target sparsity. This yields the following, which is similar to Eq. 8.

$$\min_{\boldsymbol{\theta} \in \mathbb{R}^d, \lambda_{q,i} \to \{0,1\}} \mathcal{L}_\mathcal{S}(\boldsymbol{\lambda_q} \odot \boldsymbol{\theta}) + \eta \cdot \|\boldsymbol{\lambda_q}\|_1. \tag{10}$$

Since the Gaussian KL $\phi_i$ is approximated, the distance from the initial weights is not taken into account in this setting. The authors adopt a problem setting where the weights trained a few epochs ahead instead of the initial weights are used for ticket search following Frankle et al. [11]; therefore their work does not have to consider suppressing the learning rate, i.e., the distance from the initial weights. Note that Hayou et al. [15] proposed PAC-Bayes pruning (PBP) by optimizing the PAC-Bayes bound. However, the limitation of our analysis is that we cannot reveal an explicit relationship between continuous sparsification and PBP.

## 5  Conclusion

In this work, we explored the fact that a small learning rate is required to find winning tickets, and we provided empirical analysis related to flatness and the distance from the initial weights. On the basis of these findings, we used the PAC-Bayesian framework to analyze winning tickets and experimentally showed that it captures the generalization behavior. Finally, we reconsidered IMP and continuous sparsification from a PAC-Bayesian perspective. In this study, we do not analyze the case where no solution exists near the initial weights, which needs IMP with rewinding to early epoch.

## 6  Acknowledgements

We appreciate anonymous reviewers of NeurIPS 2022 for giving constructive suggestions to improve our manuscript. IS was supported by JSPS KAKENHI Grant Number 20H05703 Japan.

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
