# A Background

In this section, we explain the background knowledge for IMP and the PAC-Bayesian theory.

**IMP** Iterative pruning is a method of obtaining a subnetwork by repeating training and pruning in stages. Frankle and Carbin [10] showed that iterative pruning (Algorithm 1) could find the winning ticket by adding an operation to restore the weights to the initial weights after pruning.

---

**Algorithm 1** Iterative Pruning for LTH

---

1: Randomly initialize the parameters $\boldsymbol{\theta}$ of a neural network to $\boldsymbol{\theta}_{\text{init}}$ and initialize a mask $\boldsymbol{m}$ to $1^{|\boldsymbol{\theta}_{\text{init}}|}$
2: Train the network $f(x; \boldsymbol{w} \odot \boldsymbol{m})$ for $T$ iterations, producing network $f(x; \boldsymbol{\theta}_T \odot \boldsymbol{m})$
3: Produce a new mask based on the current mask criterion. Rank the unmasked weights by their scores, set the mask value to 0 for the bottom $p\%$, the top $(100 - p)\%$ to 1.
4: If the target pruning ratio is satisfied, the resulting network is $f(x; \boldsymbol{\theta}_T \odot \boldsymbol{m})$
5: Otherwise, reset $\boldsymbol{\theta}$ to $\boldsymbol{\theta}_{\text{init}}$, and go back to step 2

---

In Frankle and Carbin [10], the mask criterion is simply to keep the weights with a large final magnitude, $|\theta_T|$; This is called Iterative Magnitude Pruning (IMP). This paper follows their setting: $p$ is set to 0.2 and the models are pruned globally.

Frankle et al. [11] proposed an iterative pruning with rewinding to avoid lower-stability phase to SGD noise in the early training in each IMP step. The algorithm does not return to the exact initial weights, but instead returns the weights trained slightly in advance as the initial weights. It can find the winning ticket even with the initial large learning rate or in the harder settings such as ImageNet; however it revises the original LTH setting. Our paper focuses on analyzing the properties of winning tickets rather than improving the accuracy or robustness of winning tickets; thus, we will not consider this problem setting.

Liu et al. [27] claimed that the large learning rate performs better in the larger model settings such as ResNet56 and ResNet110, contrary to Frankle and Carbin [10], which stated that IMP requires the small learning rate to find winning tickets. From our findings in Section 3.3 and 4.3, we suppose that there is no good solution near the initial weights in such a setting; therefore the small learning rate cannot find the winning tickets, and that IMP with rewinding is effective in such a setting because it shifts the initial weights closer to the final trained weights by pretraining a little and searches for the winning ticket in a better lottery.

**PAC-Bayesian Theory** First, we provide the notations used in this paper. Denote the training sample $\mathcal{S} = \{(x_i, y_i)\}_{i=1}^m \in (\mathcal{X} \times \mathcal{Y})^m$, which is randomly sampled from an underlying data distribution $\mathcal{D}$. Let $\mathcal{H}$ be a set of hypotheses, and $\ell : \mathcal{H} \times \mathcal{X} \times \mathcal{Y} \to [0, 1]$ be a loss function. Given $f \in \mathcal{H}$, we formulate the empirical risk on $\mathcal{S}$ and the generalization error on $\mathcal{D}$ as

$$\mathcal{L}_{\mathcal{S}}(f) = \frac{1}{m} \sum_{i=1}^m \ell(f, x_i, y_i); \quad \mathcal{L}_{\mathcal{D}}(f) = \mathop{\mathbb{E}}_{(x,y) \sim \mathcal{D}} \ell(f, x, y). \tag{11}$$

The PAC-Bayesian framework gives a bound on the generalization error of a posterior distribution $\mathbb{Q}$ over the hypothesis $\mathcal{H}$; we denote it $\mathcal{L}_{\mathcal{D}}(\mathbb{Q})$. It assumes that we have a prior distribution $\mathbb{P}$ on $\mathcal{H}$ which does not depend on training data, and we update it to $\mathbb{Q}$ through the learning process. Optimizing the PAC-Bayes bound controls the balance between the empirical risk and the closeness to the prior (small complexity of the model). Although there are several types of PAC-Bayes bound, we consider the following well-used form of the PAC-Bayes bound.

**Theorem 1 (Alquier et al. bound [1])** *Given a real number $\delta \in (0, 1]$, a non-negative real number $\eta$, and a prior distribution $\mathbb{P}$ on $\mathcal{H}$ defined before seeing any training sample $(X, Y) \in \mathcal{S}$, with probability at least $1 - \delta$, for all $\mathbb{Q}$ on $\mathcal{H}$*

$$\mathcal{L}_{\mathcal{D}}(\mathbb{Q}) \leq \mathcal{L}_{\mathcal{S}}(\mathbb{Q}) + \frac{1}{\eta} \left( \text{KL}[\mathbb{Q}\|\mathbb{P}] + \log \frac{1}{\delta} + \Psi(\eta, m) \right), \tag{12}$$

*where*

$$\Psi(\eta, m) \coloneqq \log \mathop{\mathbb{E}}_{\substack{h \sim \mathbb{P}, \\ \mathcal{S} \sim \mathcal{D}^m}} [\exp(\eta(\mathcal{L}_{\mathcal{D}}(h) - \mathcal{L}_{\mathcal{S}}(h)))]. \tag{13}$$

It is known that flatness is closely related to the generalization ability of neural networks [21] [20]. As shown in previous studies ([5], [31] [32]), it can be viewed in the expected sense from a PAC-Bayesian perspective. We decompose the PAC-Bayes bound as follows.

$$\mathcal{L}_{\mathcal{D}}(\mathbb{Q}) \leq \mathcal{L}_{\mathcal{S}}(f) + \underbrace{\mathcal{L}_{\mathcal{S}}(\mathbb{Q}) - \mathcal{L}_{\mathcal{S}}(f)}_{\text{expected sharpness}} + \frac{1}{\eta}\left(\underbrace{\text{KL}[\mathbb{Q}\|\mathbb{P}]}_{\text{KL}} + \log\frac{1}{\delta}\right) + \Psi(\eta, m), \qquad (14)$$

where $f \in \mathcal{H}$ is a solution obtained by a training. The expected sharpness term represents the amount of change in empirical risk around the trained weights, and the solutions in flatter minima are expected to have a relatively smaller value of this term. In the PAC-Bayesian framework, the role flatness plays in generalization behavior can be understood in this way.

For example, we consider the case where the Gaussian distribution is used for the prior and posterior. Let $\mathcal{N}(\boldsymbol{\mu}, \boldsymbol{\Sigma})$ be the Gaussian distribution, where $\boldsymbol{\mu}$ is the mean and $\boldsymbol{\Sigma}$, is the covariance matrix and $\boldsymbol{\theta}$ be the parameters of a neural network. We set a prior $\mathbb{P}$ to be $\mathcal{N}(\mathbf{0}, \sigma^2 \boldsymbol{I})$ and a posterior $\mathbb{Q}$ to be $\mathcal{N}(\boldsymbol{\theta}, \sigma^2 \boldsymbol{I})$, where $\sigma > 0$, and the KL term is calculated as $\|\boldsymbol{\theta}\|_2^2/(2\sigma^2)$. This is consistent with the conventional understanding: solutions in flat minima obtained with norm regularization can achieve good generalization accuracy.

When considering this PAC-Bayesian framework in the pruned networks, we should note that prior $\mathbb{P}$ has to be defined without depending on the training samples $\mathcal{S}$. It can be thought that if the parameter space is restricted to an unpruned weight subspace, we do not have to consider the pruned network case differently. However this is not valid because this prior depends on the structure of the pruning mask $\boldsymbol{m}$, which is found after seeing the training samples $\mathcal{S}$. Target sparsity $\beta$ does not depend on $\mathcal{S}$, so we can use it in the prior.

# B Other Experiments

## B.1 The test accuracy when SAM and NVRM are used

Table 3: Test accuracy on CIFAR10 and CIFAR100 when SAM and NVRM are used. The hyper-parameter of SAM $\rho$ is selected from $\{0.05, 0.1, 0.2, 0.5\}$, and that of NVRM $b$ is selected from $\{0.014, 0.018, 0.022, 0.026\}$. As a baseline, we show the results of SGD with a small learning rate, and the learning rate is equal for SAM and NVRM. We averaged over three times.

| Dataset | Sparsity (%) | ResNet20 | | | VGG16 | | |
|---|---|---|---|---|---|---|---|
| | | SGD | SAM | NVRM | SGD | SAM | NVRM |
| CIFAR10 | 90 | 89.1 | **89.7** | 89.3 | 91.8 | **92.9** | 92.7 |
| | 95 | **87.3** | **87.3** | 87.1 | 91.8 | **93.1** | 92.7 |
| CIFAR100 | 90 | 61.2 | **62.2** | 61.7 | 67.1 | 70.4 | **71.2** |
| | 95 | 45.2 | 45.2 | **46.7** | 66.4 | 70.5 | **71.0** |

In Section 3.2, we observe that the winning ticket is in a relatively sharper minima due to a small learning rate and that IMP can find a flatter minimum by using SAM or NVRM. Table 3 shows the test accuracy when a flatter solution is obtained by using SAM and NVRM. They can achieve a test accuracy the same as or even better than the SGD with a small learning rate, and the improvement in test accuracy can be seen especially in VGG16. It is considered that since VGG16 has a larger number of parameters than ResNet20, a flatter solution with a relatively small training loss could be found. We found no significant difference in test accuracy between SAM and NVRM.

## B.2 SAM with the large learning rate

Figure 5 shows the test accuracy when we use SAM with a large learning rate; we experimented only ResNet20 + CIFAR10 due to computational resource limitations. This figure shows that SAM does not improve test accuracy from that of IMP with a large learning rate and that SAM does not help to find winning tickets because the test accuracy continues to decline as sparsity increases. Since IMP with a large learning rate already produces relatively flatter solutions, SAM will not change the

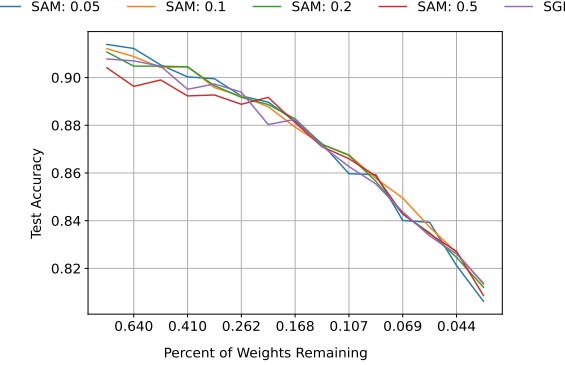

Figure 5: Test accuracy on CIFAR10 when we train ResNet20 using SAM for IMP with a large learning rate. We show the results of different SAM hyperparameters $\rho \in \{0.05, 0.1, 0.2, 0.5\}$ and SGD result as a baseline.

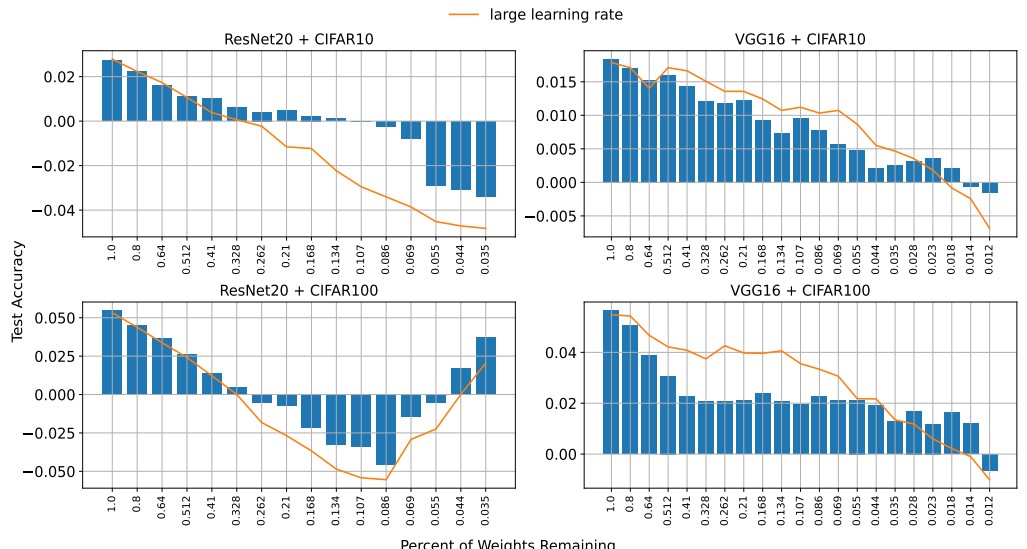

Figure 6: Test accuracy improvement on CIFAR10 and CIFAR100 when we ran IMP with a small learning rate to the target sparsity and train the subnetwork from initialization with a large learning rate (blue bar) instead of a small learning rate (0, baseline). For comparison, we plot the case of IMP with a large learning rate that corresponds to the orange line.

results as expected. This also confirms that the improved generalization accuracy when we use SAM instead of SGD for IMP with a small learning rate is because of finding flatter solution, not some other side effects of SAM.

## B.3 Mask structure with different learning rates

We conducted the following experiment related to distance from initial weights. In Figure 6, we found that IMP with different learning rates produce the pruning masks with different properties; learning with a large learning rate on a sub-network obtained with a small learning rate also does not provide winning tickets. Retraining with a large learning rate for a given mask greatly improves test accuracy at a low sparsity. In contrast, as sparsity is increased, this improvement decreases or the test accuracy worsens; it is generally the same trend as IMP with the large learning rate (orange line). We found that the mask obtained by IMP with a small learning rate has a structure that performs well with weights around the initialization, and it does not work well when trained directly of the subnetwork with a large learning rate.

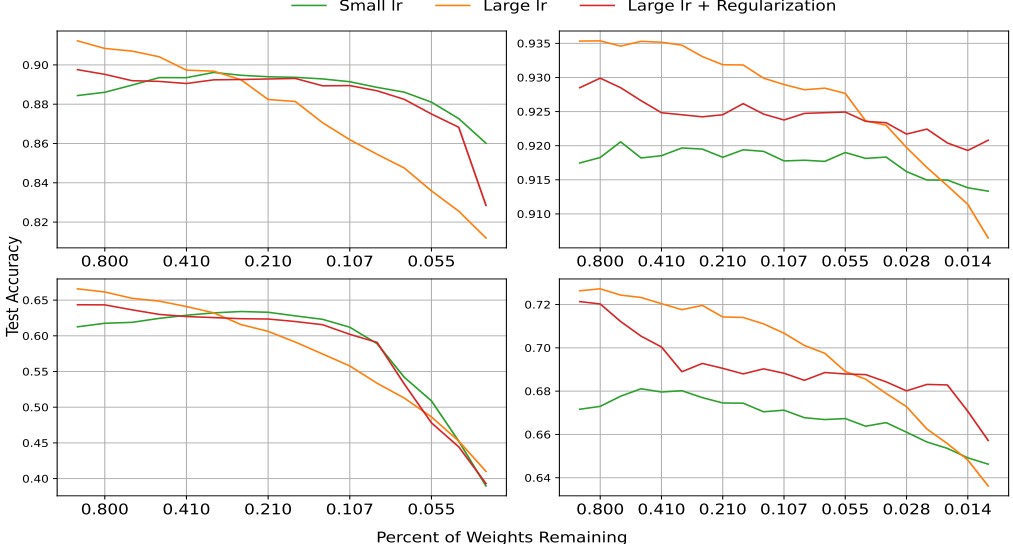

Figure 7: Test accuracy on CIFAR10 and CIFAR100 when regularization from the initial weights is added. Top Left: ResNet20 + CIFAR10, Top Right: VGG16 + CIFAR10, Bottom Left: ResNet20 + CIFAR100, Bottom Right: VGG16 + CIFAR100.

### B.4 Regularization of the distance from the initial weights

Figure 7 shows the test accuracy on CIFAR10 and CIFAR100 when we trained ResNet20 and VGG16 with a regularization from the initial weights. For a large learning rate setting (orange line), increasing sparsity significantly reduces the test accuracy, which means that IMP cannot find the winning ticket. By adding regularization from the initialization, this decrease in the test accuracy can be reduced (red line), showing a similar trend for a small learning setting (green line), which is successful in finding winning tickets.

### B.5 The distribution of parameter changing the learning rate

Figure 8 shows the parameter distribution of ResNet20 and VGG16 changing the learning rate. We trained them on CIFAR10 and plot the unpruned weights in order from the smallest to largest: $20\%$, $40\%$, $60\%$, $80\%$. The same sparsity where the winning tickets cannot be found in Figure 1 shows a change in the trend of the distribution. Although the difference is not apparent when we measure L2 norm, we confirmed that each parameter, which was near the initial weights originally, becomes distributed in a wider range when the learning rate is increased.

### B.6 Proof of Eq. 6

We estimate the deviation of $f$ when the $\Delta\boldsymbol{\theta}$ moves from the trained weights $\boldsymbol{\theta}$ using Taylor's theorem. Let $H$ be a Hessian matrix, then we have

$$
\begin{aligned}
|f(\boldsymbol{\theta} + \Delta\boldsymbol{\theta}) - f(\boldsymbol{\theta})| &= |\nabla f(\boldsymbol{\theta}) \cdot \Delta\boldsymbol{\theta} + \frac{1}{2}\Delta\boldsymbol{\theta}^\top H(\boldsymbol{\theta} + \gamma\Delta\boldsymbol{\theta})\Delta\boldsymbol{\theta}| \\
&= \frac{1}{2}\|\Delta\boldsymbol{\theta}^\top H(\boldsymbol{\theta} + \gamma\Delta\boldsymbol{\theta})\Delta\boldsymbol{\theta}\|_2 \\
&\leq \frac{1}{2}\|\Delta\boldsymbol{\theta}\|_2^2 \cdot \|H(\boldsymbol{\theta} + \gamma\Delta\boldsymbol{\theta})\|_2 \\
&\leq \frac{1}{2}\|\Delta\boldsymbol{\theta}\|_2^2 \sup_{\gamma \in [0,1]} \|H(\boldsymbol{\theta} + \gamma\Delta\boldsymbol{\theta})\|_2 \\
&= \frac{1}{2}\|\Delta\boldsymbol{\theta}\|_2^2 \sup_{\gamma \in [0,1]} \lambda_{\max}^{\boldsymbol{\theta}+\gamma\Delta\boldsymbol{\theta}},
\end{aligned}
\tag{15}
$$

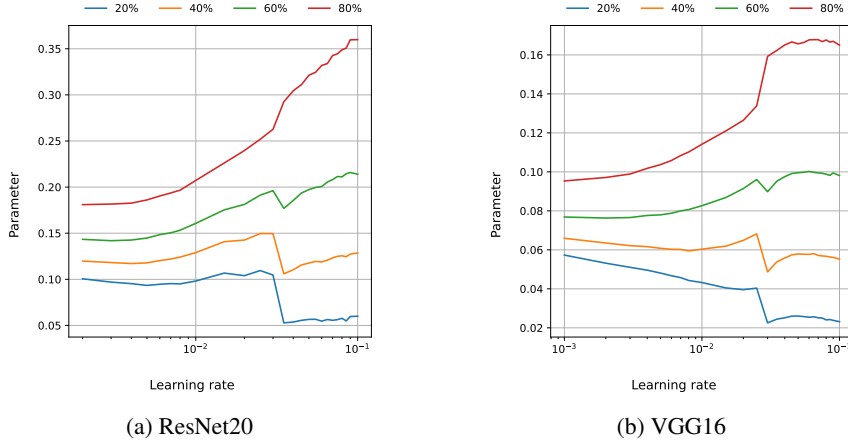

(a) ResNet20              (b) VGG16

Figure 8: Parameter distribution of ResNet20 and VGG16 trained on CIFAR10. In each learning rate, the trained parameters are plotted in the increasing order of weights: $20\%$, $40\%$, $60\%$, $80\%$. The subnetwork of ResNet20 has $90\%$ sparsity, and that of VGG16 has $99\%$ sparsity.

where $\lambda_{\max}^{\boldsymbol{\theta}+\gamma\varDelta\boldsymbol{\theta}}$ is a top eigenvalue of $H(\boldsymbol{\theta}+\varDelta\boldsymbol{\theta})$.

First equation comes from quadratic Taylor's theorem, second equation comes from the fact that $\theta$ is a trained weights and $\nabla f(\boldsymbol{\theta}) = 0$, and third inequality holds because of sub-multiplicativity of matrix norm.

## B.7 Variational KL bound

In Section 4.3, we optimized the following variational KL bound [6], [15]. Given a real number $\delta \in (0, 1]$, with probability $1 - \delta$ over the training sample $\mathcal{S}$,

$$\min \begin{cases} \mathcal{L}_{\mathcal{S}}(\mathbb{Q}) + B + \sqrt{B\left(B + 2\mathcal{L}_{\mathcal{S}}(\mathbb{Q})\right)} \\ \mathcal{L}_{\mathcal{S}}(\mathbb{Q}) + \sqrt{\frac{B}{2}} \end{cases}, \tag{16}$$

where

$$B = \frac{\mathrm{KL}(\mathbb{Q}\|\mathbb{P}) + \log\frac{2\sqrt{|\mathcal{S}|}}{\delta}}{|\mathcal{S}|}. \tag{17}$$

We used this bound in our experiment because it can avoid selecting variables that appear in the PAC-Bayes bound, such as $\eta$ in Eq. 12.