# OpenReview forum: "Analyzing Lottery Ticket Hypothesis from PAC-Bayesian Theory Perspective"
_NeurIPS.cc/2022/Conference — NeurIPS 2022 Accept_

### Official Review · Reviewer_rPEu · 2022-07-04

**Rating:** 3
**Confidence:** 2
**Soundness:** 2 fair
**Presentation:** 2 fair
**Contribution:** 2 fair

**Summary:**

This work considers the generalization ability of the winning lottery tickets in the lottery ticket hypothesis.

The first message of the paper is that the winning tickets have sharp minima and lead to worse generalization. This message is supported by the following experiments:
- Comparing the test accuracy of iterative magnitude pruning (IMP) with various learning rates with/without label noise;
- Comparing the projected 1-d landscape and calculating the trace of Hessian at convergence for IMP with small learning rate, IMP with larger learning rate, and IMP with learning rate and additional penalty terms.

The second message is that adding regularization on the deviation from the full network parameters allows finding a winning ticket with a large learning rate. This message is supported by comparing the behavior of (sub-)networks with different levels of sparsity.

The third message is that the generalization behavior of winning tickets found via IMP can be captured by a PAC-Bayesian model with a spike-and-slab prior. The choice of the spike-and-slab prior is partially justified by the second message. It is claimed that such a PAC-Bayesian model captures the behavior of winning tickets under different learning rates. This is established by comparing against the zero-mean Gaussian model on the evaluation of the risk/loss and KL-divergence for tickets found with different learning rates. For the proposed model, tickets found with large learning rates have large KL-divergences from the prior and larger risk terms than the tickets found with moderate learning rates. In section 4 the spike-and-slab model is used to provide some explanation for IMP and continuous sparsification.


**Questions:**

Please see =Strengths and Weaknesses= above.

**Strengths And Weaknesses:**

- Strength: the hyperparameters used for SAM and NVRM regularization are explicitly mentioned;
- Weakness
    + The criterion for finding the winning ticket is unclear: In the experiments reported in Figure 1, the (implied) criterion for finding the winning ticket seems to be the test accuracy being high on the clean dataset, while in Figure 3, the criterion is that the subnetwork accuracy is close to that of the full network.
    + The connection between IMP for winning tickets and the PAC-Bayesian model is not strong. The claim is solely based on the ordering of the risk and KL-divergence for three groups of tickets found through various learning rates.
    + The presentation can be improved. For example
        1. While it can be inferred that $\delta_{ \{0\} }$ stands for the delta function with a peak at zero-weight, it would be more clear to define explicitly;
        2. It is unclear how $\lambda_{p,i}$'s are chosen in the experiments.

---

> ### Author Response · Authors · 2022-08-02
> **Response to Reviewer rPEu (1/2)**
>
> Thank you for your useful question. We hope this rebuttal has answered all your questions. We pray this reply will change your decision.
>
> > Q1. (Weakness 1) The criterion for finding the winning ticket is unclear: In the experiments reported in Figure 1, the (implied) criterion for finding the winning ticket seems to be the test accuracy being high on the clean dataset, while in Figure 3, the criterion is that the subnetwork accuracy is close to that of the full network.
>
> A1. The criterion for finding the winning ticket is that whether the subnetwork is trained independently from the initial weights and shows a comparable accuracy to the original full network. This is explained in line 4 and line 27, and this criterion was commonly used in [1, 2, 3].
> While the explanation in line 102 “The subnetwork eventually performs worse than the original unpruned network, which means that IMP ultimately fails to find the winning ticket.” is correct, Figure 1 is so misleading as you pointed out, therefore we have added the unpruned baseline to the clean data setting of Figure 1 for clear criterion. For example, the table below is for ResNet20 + clean data; the full network accuracy improves even after lr = 0.03, where the subnetwork accuracy significantly deteriorates, and it is more clear that wining tickets are no longer found.
>
> | Learning rate | 0.001 | 0.005 | 0.01 | 0.03 | 0.04 | 0.06 | 0.08 | 0.1
> | ---- | ---- | ---- | ---- | ---- | ---- | ---- | ---- | ---- |
> | Test accuracy | 0.775 | 0.867 | 0.885 | **0.900** | **0.903** | **0.910** | **0.910** | **0.913**
>
>
> > Q2. (Question 2) The connection between IMP for winning tickets and the PAC-Bayesian model is not strong. The claim is solely based on the ordering of the risk and KL-divergence for three groups of tickets found through various learning rates.
>
> A2. We understood your question in two ways and will respond to each of them as follows.
>
> (1) The motivation to consider the PAC-Bayesian model in IMP is not strong. \
> The motivation to consider the PAC-Bayesian model in IMP is strong.
> In this paper, the relationship between the learning rate and IMP is the starting point, and we move to the discussion of flatness. This is because the large learning rate generally helps an optimizer to converge to flatter minima, which is advantageous for generalization [4, 5]. For flatness, it is necessary to consider not only one point in the parameter space, but also its surrounding behavior, and it is natural to consider the distribution of the parameters. In this sense, the flatness can be theoretically supported from a PAC-Bayesian perspective as shown in the previous studies [6, 7, 8]; therefore this motivates us to discuss the PAC-Bayesian theory for the winning tickets and IMP.
>
> (2) The experiment in Section 4.3 Figure 4 is not relevant and important to discuss IMP. \
> We would like to emphasize that the purpose of this experiment is quite different from the connection between IMP and the PAC-Bayes bound. In this experiment, we aim to show that the PAC-Bayes bound in our formulation is useful to analyze the winning tickets, and the emphasis is not on the algorithm IMP to find winning tickets. The connection between IMP and PAC-Bayes bound is discussed in Section 4.4.1. There we explained that IMP can be regarded as an alternating optimization of the KL and the risk based on the PAC-Bayes bound in our formulation, and we also gave experiments for confirmation in table 2.
>
> As you pointed out, the words “PAC-Bayesian theory” in line 37 suddenly appear before revised, and it is hard to understand the motivation; therefore we have revised as follows.  \
> **“In this connection, we then focus on the two concepts flatness and the distance from the inital weights of the winning tickets. We next apply the PAC-Bayesian theory to LTH on the basis of the flatness motivation and show that it can explain the relationship between LTH and generalization behavior. ”**
>
>
> [1] The Lottery Ticket Hypothesis: Finding Sparse, Trainable Neural Networks,  Frankle and Carbin, ICLR2019 \
> [2] Winning the Lottery with Continuous Sparsification, Savarese+, NeurIPS2020 \
> [3] Linear Mode Connectivity and the Lottery Ticket Hypothesis, Frankle+, ICML2020
> [4] The large learning rate phase of deep learning: the catapult mechanism, Lewkowycz+, 2020 \
> [5] A diffusion theory for deep learning dynamics: Stochastic gradient descent exponentially favors flat minima, Xie+, ICLR2021 \
> [6] Computing Nonvacuous Generalization Bounds for Deep (Stochastic) Neural Networks with Many More Parameters than Training Data, Dziugaite and Roy, UAI2017 \
> [7] Exploring generalization in deep learning, Neyshabur+, NIPS2017 \
> [8] A PAC-bayesian approach to spectrally-normalized margin bounds for neural networks, Neyshabur+, ICLR2018

---

> > ### Author Response · Authors · 2022-08-02
> > **Response to Reviewer rPEu (2/2)**
> >
> > >Q3. (Weakness 3) The presentation can be improved. For example
> > > 1. While it can be inferred that $δ_0$ stands for the delta function with a peak at zero-weight
> > > 2. it would be more clear to define explicitly It is unclear how $λ_{p, i}$’s are chosen in the experiments
> >
> > A3. Thank you for your constructive opinions.
> > 1. We have revised the paper by adding the following explanation in line 214. \
> > **“Dirac delta distribution with a peak at zero-weight $\delta_{\{0\}}$”**
> > 2. We have added the description of $λ_{p, i}$ in line 221, which is originally explained at the end of Appendix A.  \
> > Specifically,  **“$λ_{p, i}$ is set to the target sparsity, so if we want 90% sparsity winning tickets, then $λ_{p, i}$ is set to 0.1”**.

---

### Official Review · Reviewer_MZGr · 2022-07-08

**Rating:** 7
**Confidence:** 3
**Soundness:** 3 good
**Presentation:** 3 good
**Contribution:** 3 good

**Summary:**

The authors examine the generalisation of "winning tickets" trained by iterative magnitude pruning (IMP). It has been observed that a low learning rate is needed to train winning tickets, which usually leads to sharper minima being found and greater vulnerability to label noise. The authors use a series of experiments to show that the sharp minima is not a precondition for winning tickets and argue that the true condition is that the final trained weights are near to the initialisation. They show that by using different training methods, such as SAM or a larger learning rate alongside a regularisation to remain near the initialisation can also lead to winning tickets without sharp minima, which in some cases can generalise better.

They show that this behaviour can be described by a spike-and-slab PAC-Bayes formulation and use this to explain the behaviour of IMP and other methods for finding winning tickets.

**Questions:**

1. Exactly how are you motivating the use of a PAC-Bayes bound for a non-random predictor and which are you using?
2. Which of the ideas from the analysis using spike-and-slab and closeness to initialisation are new and which appear in the literature already? This affects my evaluation of the contribution of the paper significantly.

**Limitations:**

Yes I believe the paper clearly explains which problems it is addressing.

**Strengths And Weaknesses:**

The paper is relatively clear and well written. The most complex section is the empirical analysis which makes the main argument about generalisation and closeness to the initialisation, which is relatively well explained but perhaps could be prefaced by slightly more motivation for the following set of experiments (these are well explained individually but the conclusion is somewhat left to the reader). The background on PAC-Bayes and IMP is a little lacking but this may be due to space considerations.

The overall area discussed is highly interesting and of great relevance to the community.

The empirical evaluation in the paper is well thought out and reasonably comprehensive, backing up the central argument well.


**Signficance**

To my (limited) knowledge the viewpoint given of winning tickets as being a phenomenon relating to closeness to the initialisation is a new one and is quite interesting. The PAC-Bayes perspective suggested seems to have real explanatory power and represents progress in this important problem. However I am not familiar enough with the literature to say how original this explanation is.

I note though that the spike-and-slab idea itself is not new. I should also note that the authors appear to use a PAC-Bayes bound for a randomised predictor to upper bound the error of a deterministic one (i.e. $L(f) \le \mathbb{Q}_{f} L(f)$), which has been used as an assumption by some empirical works, but this should be clearly stated.


**Background**

The major caveat in my review is that it is unclear which of the most significant ideas in the paper are new, this should be more clearly explained.


----

I am happy with the answers to my questions and that the PAC-Bayes analysis has been placed on a more solid footing, so I increase my score accordingly.

---

> ### Author Response · Authors · 2022-08-02
> **Response to Reviewer MZGr**
>
> > Q1. (Question 1) Exactly how are you motivating the use of a PAC-Bayes bound for a non-random predictor and which are you using?
>
> A1. In this paper, the relationship between the learning rate and winning tickets is the starting point, and we move to the discussion of flatness. This is because the large learning rate generally helps an optimizer to converge to flatter minima, which is advantageous for generalization [1, 2]. For flatness, it is necessary to consider not only one point in the parameter space, but also its surrounding behavior, and it is natural to consider the distribution of the parameters. In this sense, the flatness can be theoretically supported from a PAC-Bayesian perspective as shown in the previous studies [3, 4, 5]; therefore this motivates us to discuss the PAC-Bayes bound for the winning tickets.
>
> As you pointed out, the words “PAC-Bayesian theory” in line 37 suddenly appear before revised, and it is hard to understand the motivation; therefore we have revised as follows. \
> **“In this connection, we then focus on the two concepts flatness and the distance from the initial weights of the winning tickets. We next apply the PAC-Bayesian theory to LTH on the basis of the flatness motivation and show that it can explain the relationship between LTH and generalization behavior. ”**
>
> The PAC-Bayes bound used in our work is called variational KL bound and is described in detail in Appendix B.7.
>
>
> > Q2. (Question 2) Which of the ideas from the analysis using spike-and-slab and closeness to initialisation are new and which appear in the literature already? This affects my evaluation of the contribution of the paper significantly
>
> A2. \
> (1) spike and slab \
> The idea of PAC-bayes bound in the context of LTH is a contribution of our work, so the idea of spike-and-slab is also new in the LTH discussion.
> There are several studies that consider spike-and-slab in the context of machine learning, and the calculation of KL divergence in this paper is based on [6]. In the context of PAC-Bayes bound, [7] adopted spike-and-slab distribution, however the motivation and the purpose are totally different from our work. We emphasized this at the end of Section 2. \
> (2) closeness to the initialization \
> In terms of LTH and closeness to the initialization, there is only [8] as far as we know. This paper refers to this distance in connection with a phenomenon called sparse double descent, and this reference is limited to a discussion when sparsity is changed. Our work focuses on the distance from the initial weights in relation to the learning rate, and actually conducts experiments by adding regularization of this distance, and furthermore, supports it from a PAC-Bayesian perspective.
> In summary, the contribution of this paper is as follows.
>  - We found that the small learning rate has a worse test accuracy under label noise than the large learning rate.
>  - As a reason for this, we confirmed that the small learning rate finds relatively sharp minima, and showed that flatter wining tickets can be obtained using SAM and NVRM.
>  - As a reason for why the small learning rate is necessary for IMP, we considered the closeness to the initial weights, and showed that a large learning rate can work well when the distance is suppressed.
>  - On the basis of these findings, we introduced the PAC-Bayes bound in the context of LTH and showed that it is consistent with our findings.
>  - We showed that existing algorithms can be seen as an optimization of PAC-Bayes bound of our formulation.
>
>
> [1] The large learning rate phase of deep learning: the catapult mechanism, Lewkowycz+, 2020 \
> [2] A diffusion theory for deep learning dynamics: Stochastic gradient descent exponentially favors flat minima, Xie+, ICLR2021 \
> [3] Computing Nonvacuous Generalization Bounds for Deep (Stochastic) Neural Networks with Many More Parameters than Training Data, Dziugaite and Roy, UAI2017 \
> [4] Exploring generalization in deep learning, Neyshabur+, NIPS2017 \
> [5] A PAC-bayesian approach to spectrally-normalized margin bounds for neural networks, Neyshabur+, ICLR2018 \
> [6] Variational Sparse Coding, Tonolini+, PMLR2020 \
> [7] Probabilistic fine-tuning of pruning masks and PAC-Bayes self-bounded learning, Hayou+, 2021 \
> [8] Can network pruning benefit deep learning under label noise?, He+, 2021（Sparse Double Descent: Where Network Pruning Aggravates Overfitting, He+, ICML2022）

---

> > ### Comment · Reviewer_MZGr · 2022-08-03
> > **response**
> >
> > Thanks for your comment, I am happy with the answer regarding originality.
> >
> > With my comment about the PAC-Bayes bound I just wanted to emphasise that the bounds in eq. 11/13 are bounds on the error of the posterior $\mathbb{Q}$, i.e. a bound on $\mathcal{L}_\mathcal{D}(\mathbb{Q})$, and not a bound on $\mathcal{L}_\mathcal{D}(f)$. Thus if you are commenting on the generalisation ability of $f$ then some additional assumption is needed. Some papers (e.g. SAM) have assumed $$$\mathcal{L}_\mathcal{D}(f) \le \mathcal{L}_\mathcal{D}(\mathbb{Q})$ in order to make statements about the generalisation of $f$ and I feel that this point needs some discussion, provided it is $f$ you wish to consider rather than $\mathbb{Q}$.
> >
> > Additionally, I now note that the PAC-Bayes bound used is not really a complete bound - the PAC-Bayes bound from Alquier et al. is not really informative without a bound on the $\Psi(\eta, m)$ term, which could easily be unbounded for the $\eta$ value chosen. Since the loss function is bounded in $[0, 1]$ you can use Hoeffding lemma to bound this term as $\eta^2 / 2m$.
> >
> > I would suggest adding an explanation of this, as otherwise the "PAC-Bayes analysis" is not really valid which I feel undermines the paper somewhat.

---

> > > ### Author Response · Authors · 2022-08-04
> > > **Re: response**
> > >
> > >  > Q1. With my comment about the PAC-Bayes bound I just wanted to emphasise that the bounds in eq. 11/13 are bounds on the error of the posterior $\mathbb{Q}$, i.e. a bound on $L_\mathcal{D}(\mathbb{Q})$, and not a bound on $L_\mathcal{D}(f)$. Thus if you are commenting on the generalisation ability of f then some additional assumption is needed. Some papers (e.g. SAM) have assumed $L_\mathcal{D} (f) \leq L_\mathcal{D}(\mathbb{Q})$ in order to make statements about the generalisation of f and I feel that this point needs some discussion, provided it is $f$ you wish to consider rather than $\mathbb{Q}$.
> > >
> > > A1. Thank you for pointing this out! We have revised to descrive this assumption in line 230. \
> > > **“Here, we make an assumption that the test error of the trained model $f$ can be bounded by the test error of a posterior distribution $\mathbb{Q}$ for the sake of an empirical analysis on the generalization behavior of $f$. ”**.
> > >
> > >  > Q2. Additionally, I now note that the PAC-Bayes bound used is not really a complete bound - the PAC-Bayes bound from Alquier et al. is not really informative without a bound on the $\Psi (\eta ,m)$ term, which could easily be unbounded for the $\eta$ value chosen. Since the loss function is bounded in [0,1] you can use Hoeffding lemma to bound this term as $\eta ^2 / 2m$.
> > >
> > > A2. Thank you for your comment! We have added the explanation about this after line 489. Specifically, \
> > > **"$\Psi (\eta ,m)$ can be bounded by the term not depending on $\mathbb{P}$, so we obtain the following bound [ _, _ ].
> > > $L_\mathcal{D}(\mathbb{Q}) \leq L_\mathcal{S}(\mathbb{Q}) + \frac{1}{\eta} (\operatorname{KL}[\mathbb{Q} \| \mathbb{P}] + \log \frac{1}{\delta}) + \frac{η}{2m}$"**
> > >
> > > **[_] On the properties of variational approximations of Gibbs posteriors, Alquier+, JMLR2016** \
> > > **[_] PAC-Bayesian Theory Meets Bayesian Inference, Germain+, NIPS2016**

---

> > > > ### Comment · Reviewer_MZGr · 2022-08-04
> > > > **response**
> > > >
> > > > Thanks, I am happy that this part of the theoretical analysis is now clear.

---

> > > > > ### Author Response · Authors · 2022-08-08
> > > > > **Re: response**
> > > > >
> > > > > Thank you for your reply.
> > > > >
> > > > > > This affects my evaluation of the contribution of the paper significantly.
> > > > >
> > > > > I hope our response positively affects your evaluation!

---

### Official Review · Reviewer_RxB8 · 2022-07-12

**Rating:** 7
**Confidence:** 5
**Soundness:** 3 good
**Presentation:** 4 excellent
**Contribution:** 3 good

**Summary:**

The paper studies how the scale of the learning rate and properties such as $\ell_2$ regularization and distance to initialization can affect the performance of sparse subnetworks, especially in the setting of trying to extract winning tickets. Empirical results show that the distance to initialization can play an important role when searching for winning tickets, and regularizing such distance leads to better performing tickets when training with large learning rates. The PAC-Bayesian framework is used to validate some of the presented ideas and to draw connections between IMP, CS, and a minimization of a PAC-Bayes bound using spike and slab distributions.

**Questions:**

It would be extremely valuable if the ticket depicted in the top left plot of Fig. 3 could achieve a performance similar to a dense model when trained with standard settings, which seems to achieve around 92% test accuracy.

One question is whether the authors have tried to adopt the l2_init regularizer with $\lambda \approx 5 \cdot 10^{-4}$ (the value at which the orange curve 'spikes' in the top left plot) when training the dense network to extract the binary mask, but then switching to l2_norm and a different value for $\lambda$ when re-training the sparse subnetwork -- and if yes, was there any performance improvement compared to adopting the same $\lambda$ and the l2_init regularizer during both phases?

My second question is whether the authors have tried to combine the l2_init regularizer with CS instead of IMP, and whether the same results were observed (orange curve doesn't get close to ~92%).

**Limitations:**

The main limitation is described in the last point of 'Strengths And Weaknesses', more specifically the fact that the ticket fails to achieve a performance similar to the dense model trained with standard $\ell_2$ regularization and $\lambda \approx 10^{-4}$.

**Strengths And Weaknesses:**

Strengths:

- The paper is clear and has a clean presentation.

- The findings are interesting, especially how regularizing the distance to initialization can be beneficial when searching for winning tickets even with large learning rates. I think this is promising since there is a significant advantage in having a sparse subnetwork that can be trained from scratch compared to one that can only be trained from an 'early iterate' (as is commonly the case for networks like ResNets when trained with large learning rates): in this case the winning ticket can be completely defined by the binary mask and the random seed used for initialization, instead of having to store/transmit all the parameter values (as in the case of subnets that can only be re-trained from an early epoch).

- The PAC-Bayesian perspective is also interesting, especially Fig. 4 where the two main terms are shown for subnetworks extracted with different learning rates, showing that the KL divergence is indeed smaller for winning tickets.

- Lastly, the connections drawn between minimizing the PAC-Bayes bound and the two ticket search methods IMP and CS are valuable and might have practical implications on how to improve ticket search.



Weaknesses:

- The experimental results are constrained to the CIFAR 10/100 datasets. While ticket search on ImageNet can be quite costly, it is hard to draw final conclusions from results only on CIFAR.

- While the observations regarding learning rate x label noise (Sec 3.1) and learning rate x sharpness (Sec 3.2) are valuable, they seem a bit disconnected from the rest of the paper. While there is some inherent connection between the sharpness observations and the PAC-Bayesian analysis since the PAC-Bayes bound itself has its own measure of sharpness, the learning rate x label noise seems somewhat out of context compared to the rest of the paper. Additional discussion connecting Sections 3.1, 3.2, 3.3 and 4 would be helpful here, or either moving Section 3.1 to the end of the paper (or even to the Appendix) in case there isn't a strong connection between it and the main ideas.

- The biggest weakness, in my opinion, is regarding the results shown in Figure 3. The paper claims that regularizing the distance to init (l2_init) enables one to find winning tickets even when adopting large learning rates, and although in the top left plot of Fig. 3 we do see the 90% sparse ticket performing similarly to the dense network, this considers a suboptimal regularization strength for the dense model. More specifically, the dense model seems to achieve ~92% test acc. with standard $\ell_2$ regularization and $\lambda \approx 10^{-4}$ (top right plot), but in the setting where paper claims to find a winning ticket with large learning rate (top left plot, $\lambda \approx 5 \cdot 10^{-4}$), the dense model seems to yield around 88% test accuracy. Ideally, one would want to achieve close to 92% test acc. when re-training the subnetwork from scratch.

---

> ### Author Response · Authors · 2022-08-02
> **Response to Reviewer RxB8 (1/2)**
>
> Thank you for the review. We appreciate your thorough considerations on improving the work!
>
> > Q1. (Question, Weakness 3) It would be extremely valuable if the ticket depicted in the top left plot of Fig. 3 could achieve a performance similar to a dense model when trained with standard settings, which seems to achieve around 92% test accuracy.
>
> A1. We suppose that this subnetwork could NOT achieve around 92% accuracy because the winning ticket around these high sparsities have not achieved it in the previous research.
>
> Note that Figure 3 has the regularization parameter λ on the x-axis and not the sparsity, which is common in winning tickets experiments, therefore only the 90% and 95% sparsities are picked in Figure 3. The results of changing the sparsity under the appropriate λ are in Appendix B.4 Figure 8 (ResNet20+CIFAR10 is in top left), the large learning rate (lr = 0.1) with regularization (red line) can behave similarly to the small learning rate (lr = 0.01). In Figure 8, your question is that the found subnetwork may be comparable to the full network under the same regularization condition (left end of red line, about 90%), but it is not comparable to the full network in better conditions (large learning rate, left end of orange line, about 92%).
>
> LTH original paper [1] addressed this issue by adjusting the learning rate and adopted lr = 0.03 + warmup because a small learning rate has worse performance of the full network; they showed this subnetwork is comparable to the full network with a large learning rate. It is important to note that in Figure 8 of [1], the accuracy is comparable to that of full network in the sparsity range of 60%~80%, and at 90% sparsity, the accuracy is worse than that, and at 95% sparsity, the accuracy is rather worse than lr = 0.01. Therefore, even in the LTH original paper, the sparsity in our paper Figure 3 does not record an accuracy comparable to the full network with the large learning rate. In addition, it might be effective to adjust λ during training that corresponds to dynamic adjustment of the learning rate like warmup, but we have not done this. Furthermore, due to computational resource constraints, the experiments in Figure 3 were only conducted in λ = $1.0 \cdot 10^{-6}, 5.0 \cdot 10^{-6}, 1.0 \cdot 10^{-5}, …, $ so a more intensive parameter search could improve this accuracy even further.
>
> > Q2. (Question 1) One question is whether the authors have tried to adopt the l2_init regularizer with λ ≈ $5 \cdot 10^{-4}$ (the value at which the orange curve 'spikes' in the top left plot) when training the dense network to extract the binary mask, but then switching to l2_norm and a different value for λ when re-training the sparse subnetwork -- and if yes, was there any performance improvement compared to adopting the same λ and the l2_init regularizer during both phases?
>
> A2. The answer is NO, we can show the experiment in Appendix B.3 Figure 6. This experiment was designed to test whether the training with the large learning rate after getting the mask structure using IMP with the small learning rate leads to better accuracy; this is based on the fact the small learning rate is advantageous for IMP and the large learning rate is advantageous for generalization ability. Figure 6 shows that this switch can improve accuracy at low sparsity, but as sparsity increases, the accuracy deteriorates rather than improves. Compared to IMP with the large learning rate (orange line), it is clear that this switching does not work well.
> From Appendix B.4 Figure 7, we can see that the large learning rate + l2_init and the small learning rate behave similarly.  We can conclude that the solution found by IMP with the large learning rate + l2_init has a structure that works well near the initial weights, and switching to the large learning rate will fail.
>
> > Q3. (Question 2) My second question is whether the authors have tried to combine the l2_init regularizer with CS instead of IMP, and whether the same results were observed (orange curve doesn't get close to ~92%).
>
> A3. We do not analyze the CS results for the following reasons. Please note that the CS original paper [2] adopted weight rewinding setting (2 epoch) in the VGG16, ResNet20 experiments (Figure 1 in [2]). It is not known whether the large learning rates does not work well when CS applies the output mask to the exact initial values. Therefore, CS for the exact initial weights would require a comprehensive investigation like our work for IMP, and considering the hyperparameter search in Figure 3, CS with l2_init experiment was not done in our work from a computationally limitation. Thank you for this interesting point, and we would like to leave it as a future work.

---

> > ### Author Response · Authors · 2022-08-02
> > **Response to Reviewer RxB8 (2/2)**
> >
> > > Q4. (Weakness 1) The experimental results are constrained to the CIFAR 10/100 datasets. While ticket search on ImageNet can be quite costly, it is hard to draw final conclusions from results only on CIFAR.
> >
> > A4. It is known that finding winning tickets at ImageNet scale does not work well with IMP that rewinds to the exact initial weights, so IMP that rewinds to slightly trained weights is used [3]. In order to analyze in such a situation, it is necessary to consider a data-dependent PAC-Bayes bound because the prior mean is set to this rewinding weights and it is dependent on the training data. It is a different situation from our work, therefore we have not analyzed such a setting as mentioned at the end of Section 5.
> >
> > > Q5. (Weakness 2) While the observations regarding learning rate x label noise (Sec 3.1) and learning rate x sharpness (Sec 3.2) are valuable, they seem a bit disconnected from the rest of the paper. While there is some inherent connection between the sharpness observations and the PAC-Bayesian analysis since the PAC-Bayes bound itself has its own measure of sharpness, the learning rate x label noise seems somewhat out of context compared to the rest of the paper. Additional discussion connecting Sections 3.1, 3.2, 3.3 and 4 would be helpful here, or either moving Section 3.1 to the end of the paper (or even to the Appendix) in case there isn't a strong connection between it and the main ideas.
> >
> > A5. Thank you for your constructive comment! To clarify the connection, we have revised line 87 ~ line 95 as follows. \
> > **“The small learning rate is good for finding the winning ticket, which is contrary to the intuition that a large learning rate is good in terms of generalization ability. First, we show the small learning rate is actually disadvantageous for generalization ability of subnetworks through experiments under label noise. Next, as a reason for this, we show that the small learning rate finds the relatively sharper minima, and it is possible to find the winning tickets in the flatter minima using sharpness regularization. We also focused on the distance from the initial weights as a reason for why the small learning rate is important for IMP. On the basis of these findings, further discussion is developed to theoretically support them in the next section.”** \
> > We would like to leave Section 3.1 in the main text because this experiment and Figure 1 motivates our work.
> >
> > [1] The Lottery Ticket Hypothesis: Finding Sparse, Trainable Neural Networks,  Frankle and Carbin, ICLR2019 \
> > [2] Winning the Lottery with Continuous Sparsification, Savarese+, NeurIPS2020 \
> > [3] Linear Mode Connectivity and the Lottery Ticket Hypothesis, Frankle+, ICML2020

---

> > > ### Comment · Reviewer_RxB8 · 2022-08-09
> > > **Response**
> > >
> > > Thanks for the detailed response.
> > >
> > > My only suggestion would be for the authors to add a brief discussion of the additional experiments in Appendix B.3 and Appendix B.4 to the main paper. More specifically, line 161 of the paper contains 'Other settings in Appendix B.4' in parentheses, but I think going more in detail here would be valuable since the experiments in B.3 and B.4 are really interesting but easy for a reader to overlook since they are not currently detailed in the main text.
> > >
> > > It would also be interesting to try more values for $\lambda$ and/or different sparsities (lower than 90%) for Figure 3, as I think it would be extremely valuable if the gap between the winning ticket and ~92% accuracy achieved by the dense model (with optimal hyperparams) were to be diminished. I agree with that authors that achieving 92% when training a very sparse ticket from scratch seems unlikely, but perhaps this gap can be further decreased with a different value for $\lambda$ or by slightly decreasing the ticket's sparsity. These are just general suggestions that I think could further improve the paper, but that I don't see as being required or crucial changes.

---

> > > > ### Author Response · Authors · 2022-08-09
> > > > **Re: Response**
> > > >
> > > > > Q1. My only suggestion would be for the authors to add a brief discussion of the additional experiments in Appendix B.3 and Appendix B.4 to the main paper. More specifically, line 161 of the paper contains 'Other settings in Appendix B.4' in parentheses, but I think going more in detail here would be valuable since the experiments in B.3 and B.4 are really interesting but easy for a reader to overlook since they are not currently detailed in the main text.
> > > >
> > > > A1. Thank you for your reply! We have revised as follows.  \
> > > > (Before) \
> > > > **(Other settings in Appendix B.4.)** \
> > > > (After) \
> > > > **This finding is shown more clearly in Appendix B.4.  We confirmed this by examining the test accuracy when sparsity is changed for problem settings other than ResNet20+CIFAR10.**
> > > >
> > > > We have added the folloing explanation after line 148. \
> > > > **In addition, we discuss the mask structure in Appendix B.3 related to the distance from the initial weights.**
> > > >
> > > >  > Q2. It would also be interesting to try more values for $\lambda$ and/or different sparsities (lower than 90%) for Figure 3, as I think it would be extremely valuable if the gap between the winning ticket and ~92% accuracy achieved by the dense model (with optimal hyperparams) were to be diminished. I agree with that authors that achieving 92% when training a very sparse ticket from scratch seems unlikely, but perhaps this gap can be further decreased with a different value for $\lambda$ or by slightly decreasing the ticket's sparsity. These are just general suggestions that I think could further improve the paper, but that I don't see as being required or crucial changes.
> > > >
> > > > A2. Thank you for your constructive comment! \
> > > > We are now running these experimental setting with more $\lambda$ and smaller sparsities to investigate if we can improve the accuracy and have a plan to revice the current paper.

---

> > > > > ### Comment · Reviewer_RxB8 · 2022-08-09
> > > > > **Response**
> > > > >
> > > > > Thanks for the quick reply.
> > > > >
> > > > > I believe these revisions will be very valuable since the Appendix has interesting experiments which I myself hadn't given enough attention when reading the paper for my original review. Although some of the results in Appendix B.2, B.3 and B.4 are negative (as in, they show different experiments that try to extract winning tickets or improve tickets' performance/sparsity but ultimately aren't successful), I think they are valuable and of interest to the LTH / ticket search community.
> > > > >
> > > > > I have increased my score accordingly to reflect this and thank the author for the revisions.

---

### Official Review · Reviewer_K8ue · 2022-07-21

**Rating:** 8
**Confidence:** 5
**Soundness:** 4 excellent
**Presentation:** 4 excellent
**Contribution:** 4 excellent

**Summary:**

This work analyzes the key factors or indicators behind the successful identification of winning tickets in Lottery Ticket Hypothesis (LTH). That is, the flatness of the trained models and the closeness of the trained weights to the initialization. The motivation is that finding winning tickets requires small learning rates. How large a learning rate we use is related to the above two indicators. Empirical results turn out to show that the flatness seems to be of less relevance because we can find winning tickets either way: (1) using a small learning rates, which leads to winning tickets with sharp minima; or (2) using a large learning with sharpness regularization, which leads to winning tickets with flat minima.

The authors then empirically verified that properly regularizing the distance between the trained weights and the initial weights helps to find winning tickets when using a large learning rate. And they provide a framework to analyze this from the PAC-Bayesian perspective by setting the prior mean as the initial weights. Then IMP can be interpreted as alternating between minimizing the training risk term and the KL term in PAC-Bayesian framework.

**Questions:**

See the weaknesses.

**Limitations:**

The limitations are addressed well.

**Strengths And Weaknesses:**

Strengths:

+ The authors designed a delicate series of experiments to dissolve the implication of flatness and to show that the weight closeness to the initialization seems to be a more related factor.
+ The PAC-Bayesian perspective framework is a novel and interesting viewpoint and the resulting interpretation of IMP is also interesting and convincing.
+ The paper is overall clearly presented and will be of interest for most of researchers in the sparse neural network community.

----------------------------------------

Weaknesses:

- One constructive feedback I could give is that some presentations of plots could be improved for clearer comparisons. For example, I think the unpruned baselines could be added to plots in Figure 1. And to show that the closeness regularization really helps to find winning tickets, it would be better to compare large learning rates plus $\ell_2$_init with unpruned baselines with small learning rates.
- I think it would be nice to know, if flatness is not the key factor behind winning tickets, whether the sharpness regularization like SAM happens to have an implicit regularization on the weight closeness so that it also helps find winning tickets.
- The authors mentioned in the conclusion that they didn't "analyze the case 307 where no solution exists near the initial weights, which needs IMP with rewinding to early epoch." In my opinion, the failure of finding winning tickets in large networks (which is the motivation to introduce weight rewinding in Frankel's work) can also be explained by the trade-off between the minimization of risk term and KL term under the PAC-Bayesian framework --- because training larger networks makes the trade-off so hard that the alternative minimization with IMP also fails. I wonder if the authors have tried this but got no improvement or have not tried at all.

---

> ### Author Response · Authors · 2022-08-02
> **Response to Reviewer K8ue (1/2)**
>
> Thank you for the review. We highly appreciate that you positively evaluate our work!
>
> > Q1. (Weakness 1) One constructive feedback I could give is that some presentations of plots could be improved for clearer comparisons. For example, I think the unpruned baselines could be added to plots in Figure 1.
>
> A1. Thank you for your constructive comment. We have added unpruned baseline to the clean data setting in Figure 1 for clear criterion for finding winning tickets. For example, the table below is for ResNet20 + clean data; the full network accuracy improves even after lr = 0.03, where the subnetwork accuracy significantly deteriorates, and it is more clear that wining tickets are no longer found.
>
> | Learning rate | 0.001 | 0.005 | 0.01 | 0.03 | 0.04 | 0.06 | 0.08 | 0.1
> | ---- | ---- | ---- | ---- | ---- | ---- | ---- | ---- | ---- |
> | Test accuracy | 0.775 | 0.867 | 0.885 | **0.900** | **0.903** | **0.910** | **0.910** | **0.913**
>
> > Q2. (Weakness 2) And to show that the closeness regularization really helps to find winning tickets, it would be better to compare large learning rates plus ℓ2_init with unpruned baselines with small learning rates.
>
> A2. We have added the horizontal line that represents the unpruned baseline with small learning rate to the top left of Figure 3. The table below shows the unpruned baseline with the small learning rate and the test accuracy of the 90% sparsity subnetwork with the large learning rate + l2_init (orange line).
>
> | λ | (baseline) | ... | $1.0 \cdot 10^{-5}$ | $5.0 \cdot 10^{-5}$ |  $1.0 \cdot 10^{-4}$ | $5.0 \cdot 10^{-4}$ | $1.0 \cdot 10^{-3}$ | $5.0 \cdot 10^{-3}$ | ... |
> | ---- | ---- | ---- | ---- | ---- | ---- | ---- | ---- | ---- | ---- |
> | Test accuracy | **0.886** | ... | 0.857| 0.863 | 0.873| **0.883** | 0.861| 0.747 | ... |
>
> Note that the accuracy of 95% sparsity is below the full network baseline, but this is also true for the LTH original paper [1] (Section 4, Figure 8). In Appendix B.4 Figure7, we showed the test accuracy with l2_init when sparsity is changed as a comparison with the small learning rate baseline; this may also be helpful!
>
> > Q3. (Weakness 3) I think it would be nice to know, if flatness is not the key factor behind winning tickets, whether the sharpness regularization like SAM happens to have an implicit regularization on the weight closeness so that it also helps find winning tickets.
>
> A3. This is a good point. The answer is NO; the sharpness regularization like SAM does not play a role in helping to find the winning tickets by implicitly regularizing the distance from the initial weights. In Appendix B.2, we investigated whether IMP with the large learning rate would be able to find the winning tickets by using SAM. Figure 5 shows there is no significant change in the behavior, and the winning tickets are still not found.
> In summary, we showed IMP with the small learning rate finds relatively sharper minima, however sharp minima are not important for the winning tickets, and IMP with the small learning rate can find winning tickets in flatter minima whose accuracy does not degrade from the original by using SAM and NVRM. If IMP cannot find the winning tickets originally, then using these optimizers is not effective to find them.

---

> > ### Author Response · Authors · 2022-08-02
> > **Response to Reviewer K8ue (2/2)**
> >
> > > Q4. (Weakness 4) The authors mentioned in the conclusion that they didn't "analyze the case 307 where no solution exists near the initial weights, which needs IMP with rewinding to early epoch." In my opinion, the failure of finding winning tickets in large networks (which is the motivation to introduce weight rewinding in Frankel's work) can also be explained by the trade-off between the minimization of risk term and KL term under the PAC-Bayesian framework --- because training larger networks makes the trade-off so hard that the alternative minimization with IMP also fails. I wonder if the authors have tried this but got no improvement or have not tried at all.
> >
> > A4. Thank you for your interesting and valuable perspective! The table below shows the drop of training accuracy when we use the large learning rate, where we need IMP with weight rewinding to avoid considering learning rate constraints. The accuracy drop after retraining has become larger compared to table 2 in our paper (It is also interesting that the drop after pruning is very small, but this is not important here. ).  In settings that require IMP with weight rewinding, the training in the early epoch is not stable to SGD noise as shown in [2], and the pruned weights may not return to their original accuracy well when retrained as in the experiments of table2.
> > | Learning rate | After pruning (%)  | After rewinding (%) | After retraining (%) |
> > | ---- | ---- | ---- | ---- |
> > | large learning rate | -0.40 | -89.3 | **-4.4** |
> >
> > Regarding this question, we also suppose that the reason why IMP with weight rewiding works well can be considered that it is still consistent with the optimization of PAC-Bayes bound by using slightly trained weights as a prior mean. However, since prior depends on the data, we need to consider a data-dependent bound. This is a different situation, therefore we have left it to the future work.
> >
> > [1] The Lottery Ticket Hypothesis: Finding Sparse, Trainable Neural Networks,  Frankle and Carbin, ICLR2019　\
> > [2] Linear Mode Connectivity and the Lottery Ticket Hypothesis, Frankle+, ICML2020

---

> > ### Comment · Reviewer_K8ue · 2022-08-09
> > **Thanks for the detailed response**
> >
> > My questions were addressed well by the detailed response from the authors.

---

### Meta-Review · Area_Chair_XBVA · 2022-08-23

**Recommendation:** Accept
**Confidence:** Certain

**Metareview:**

Overall: The paper analyzes the key factors or indicators behind the successful identification of winning tickets in Lottery Ticket Hypothesis (LTH).

Reviews: The paper received four reviews. Strong accept (absolutely confident), Accept (Absolutely confident), Accept (confident) and Reject (less confident). It seems that there are at least three reviewer that will champion the paper for publication. The reviewers
found the paper is clear and has a clean presentation. The findings are interesting, as well as the PAC-Bayesian perspective. The authors have provided extensive answers to reviewers' comments, answering most of them successfully.

Main issues raised by reviewers:
- The criterion for finding the winning ticket is unclear
- The connection between IMP for winning tickets and the PAC-Bayesian model is not strong
- The presentation can be improved.
However, the authors have tackled most of the concerns raised.

After rebuttal: The authors have provide extensive clarifications with many additional experimental results, addressing several of the issues raised by the reviewers (some of them acknowledged this effort and they have raised their scores). Overall, the engaged reviewers seem happy with the changes and propose acceptance.

Confidence of reviews: Overall, the reviewers are fairly confident. We will put more weight to the reviews that got engaged in the rebuttal discussion period.

**Award:**

No

---

### Decision · Program_Chairs · 2022-09-14

Accept